# Smoking Cessation Advisors’ Perspectives on Pregnant Women’s Attitudes on the Risks and Consequences of Smoking While Pregnant

**DOI:** 10.3390/healthcare13233018

**Published:** 2025-11-22

**Authors:** Iliatha Papachristou Nadal, Penny Xanthopoulou, Michael Ussher

**Affiliations:** 1Division of Care in Long-Term Conditions, King’s College London, James Clerk Maxwell Building, 57 Waterloo Rd, London SE1 8WA, UK; 2Department of Medicine, Nursing and Allied Health Professionals, University of Exeter, St Luke’s Campus, Heavitree Road, Exeter EX1 2LU, UK; p.d.xanthopoulou@exeter.ac.uk; 3Division of Community Health Sciences, St George’s University of London, Cranmer Terrace, London SW17 0RE, UK; mussher@sgul.ac.uk

**Keywords:** smoking cessation, pregnancy, women’s health, behavioural medicine, health promotion, qualitative research

## Abstract

Smoking while pregnant is known to increase infant mortality, morbidity and can have adverse effects on children’s physical and mental development. Despite strong evidence and public health efforts, cessation support for pregnant smokers remains inconsistent across UK maternity services. **Objectives:** To investigate smoking cessation advisors’ perspectives on pregnant women’s attitudes and beliefs about smoking during pregnancy and the perceived risks to the unborn child. **Methods:** Ten smoking cessation advisors, from ten different hospital sites in London and Kent, participated in a focus group. Data were audio-recorded, transcribed verbatim, and thematically analysed. **Results:** Three main themes were identified: (1) building a positive relationship, (2) pregnant smokers’ awareness of health risks, and (3) informative techniques used by smoking cessation advisors. While based on a single focus group, the findings suggest that while many pregnant smokers recognise the health risks, this awareness alone may not lead to cessation. **Conclusions:** Adopting a person-centred approach that considers pregnant smokers’ knowledge, beliefs, and emotional readiness may help advisors deliver timely and appropriate information to encourage cessation. Practice implications could benefit from a more structured yet flexible framework to guide sensitive discussions about smoking risks and to reinforce the potential benefits of quitting for both maternal and foetal wellbeing.

## 1. Introduction

Maternal smoking remains one of the leading preventable causes of adverse pregnancy outcomes [1]. Recent meta-analytic and national cohort studies confirm that prenatal smoking increases the risk of miscarriage and preterm birth by 30–40%, doubles the likelihood of low birth weight, and raises the risk of stillbirth by about 50% [2,3]. Smoking during pregnancy has also been associated with metabolic complications, although recent evidence shows no clear relationship with gestational diabetes (OR 1.06, 95% CI 0.95–1.19) [4,5]. In the United Kingdom, 9–10% of women report smoking during pregnancy, rising to over 15% in socioeconomically deprived populations [6,7]. Prenatal exposure also increases the risk of sudden infant death syndrome and long-term developmental and behavioural problems in children [8].

Despite decades of public health campaigns, smoking during pregnancy continues to contribute substantially to perinatal morbidity and mortality, particularly among disadvantaged women [6]. Financial stress and social disadvantage continue to shape smoking behaviours and reduce cessation success [9].

Recent systematic reviews and meta-analyses show that although most pregnant smokers understand the risks of smoking, knowledge alone rarely leads to behaviour change [10,11]. Psychological dependence, social norms, stigma, and the normalisation of risk frequently undermine cessation efforts, even among women who acknowledge harm to the unborn child [12,13,14,15]. Those who successfully quit tend to hold stronger beliefs about the severity of smoking-related harm, though the strength and direction of this association remain uncertain [16].

Health professionals play a pivotal role in supporting pregnant smokers to stop, yet evidence shows that many lack confidence, time, or adequate training to deliver consistent advice [17]. In response, the National Institute for Health and Care Excellence (NICE, 2025) guidelines emphasise personalised, non-judgemental communication based on behavioural science principles [18]. Previous work by Campbell et al. (2018) identified a range of effective pregnancy-specific behaviour change techniques (BCTs) [19]. This includes goal setting, feedback, and risk communication. However, these strategies are not consistently applied in practice. Tailoring interventions to women’s individual beliefs, emotions, and social contexts remains essential. Recent evidence by Khanal et al., 2023, further highlights the value of social support; for instance, partner involvement significantly enhances cessation success during pregnancy and postpartum [20].

Although the 2023 National Centre for Smoking Cessation and Training (NCSCT) guidelines outline best practice, consistent implementation remains limited [21]. Midwives and smoking cessation professionals frequently describe challenges in building rapport with pregnant smokers and tailoring interventions to individual beliefs and circumstances [17]. Understanding how smoking cessation advisors (SCAs) perceive pregnant women’s attitudes and beliefs about smoking may help strengthen communication strategies and inform the development of more responsive intervention frameworks. This study, therefore, explores SCAs’ perspectives on pregnant women’s views on smoking during pregnancy and the perceived risks to the unborn child.

## 2. Materials and Methods

### 2.1. Participants and Recruitment

This study consisted of ten experienced, non-smoking female smoking cessation advisors (SCAs), including four research nurses, five midwives, and one health psychologist. All SCAs were trained to the NCSCT standards [21]. Participants were recruited using purpose sampling from the London Exercise and Pregnant Smokers (LEAP) randomised controlled trial (RCT). This trial was conducted between April 2009 and January 2014, and evaluated whether adding a physical-activity intervention (mainly walking on a treadmill) to standard cessation support improved quit rates among pregnant smokers [22,23]. Each SCA was based at one of ten hospitals across London and Kent: St Mary’s Hospital, Chelsea and Westminster Hospital, Epsom and St Helier University Hospitals, St George’s Hospital, Kingston Hospital, Guy’s and St Thomas’s, Croydon University Hospital, Crawley Hospital, Medway Hospital, and King’s College Hospital.

In the main trial, face-to-face data collection with pregnant participants was conducted by these trained SCAs. The SCAs delivered weekly behavioural cessation sessions and recorded smoking status and intervention data at each visit. Those eligible for recruitment were pregnant women between 10 and 24 weeks’ gestation; smoked five or more cigarettes per day before pregnancy and at least one per day during pregnancy. This gestational range ensured enrolment early enough in pregnancy for smoking cessation to benefit foetal development. The smoking thresholds identified women with established, regular smoking patterns suitable for behavioural intervention [22,23]. Exclusion criteria included medical or obstetric complications that could limit physical activity, inability to complete questionnaires in English, substance dependence, or planned use of nicotine replacement therapy at baseline. The main RCT employed intention-to-treat analysis with logistic regression to compare smoking cessation outcomes between intervention groups [23]. For this study, the inclusion criteria required that participants were actively providing or had recently provided smoking cessation support to pregnant women within NHS hospitals or community maternity services and had completed NCSCT-accredited training. Exclusion criteria included advisors without direct client contact or those not trained to the national standard.

These participants were the main SCA for pregnant women in their allocated hospital and had contact with up to 78 pregnant smokers over the duration of the trial (A total of 789 pregnant smokers participated in the overall trial [23]. The focus group for this qualitative study was conducted in June 2012, during the third year of the LEAP trial.

All the advisors had regular meetings with their clients, offering six weekly sessions of 20 min of individual behavioural cessation support. The SCA supported smoking cessation by reinforcing commitment to abstinence and solving women’s problems about maintaining abstinence. Pregnant women were randomised to either behavioural cessation support alone or behavioural cessation support plus a physical activity intervention (14 sessions of supervised exercise, walking on a treadmill for up to 30 min with physical activity consultations). The physical activity support aimed to identify opportunities and motivate the women to include physical activity in their lives, thus reducing the urge to smoke.

The last author (MU), a male professor of behavioural science and principal investigator of the trial, invited SCAs to participate via email. All had prior professional contact with him through the ongoing study. The focus group took place in June 2012, during year three of the trial. Invitation emails included an information sheet and consent form, and participants were given three weeks to respond.

This study was approved by the Wandsworth Research Ethics Committee (reference number 08/H0803/177). All participants provided written informed consent.

### 2.2. Study Design

Following the Consolidated Criteria for Reporting Qualitative Research (COREQ), a face-to-face focus group was conducted [24], Appendix A. A focus group approach was chosen to facilitate discussion and shared reflection among advisors, allowing participants to build on each other’s experiences and provide a collective understanding of professional practice. [25]. A focus group is a strong and valuable method that allows for a group perspective to develop and refine health education messages and interventions [26].

### 2.3. Topic Guide and Procedures

A topic guide was developed using open-ended questions to understand the SCAs’ perspectives on pregnant women’s attitudes on the risks and consequences of smoking while pregnant. The topic guide was developed collaboratively by the study team, drawing on prior literature and practical experience of smoking cessation support, and piloted informally with one advisor and the last author (MU) to ensure clarity and relevance. The topic guide covered the following questions of interest (complete topic guide Appendix A):What are the opinions of clients on smoking while pregnant? Do clients really think that smoking harms their unborn baby?What is the advisor’s perception of discussing the risks and consequences of smoking while pregnant? Is it important to raise these issues?How as an advisor does one deal with this?What methods and tactics are used?

The moderator (last author), who also served as the principal investigator of the main trial, facilitated the focus group. While their broader involvement in the project could have introduced bias, they were not directly engaged in participant support or smoking cessation delivery, thereby reducing the likelihood of influencing responses. An independent research administrator took field notes of the focus group. This provided background information for the transcription and analysis. The focus group was conducted in a quiet meeting room at St George’s University of London. The data from this focus group, comprising 90 min and was recorded using a digital audio recorder and transcribed verbatim.

### 2.4. Data Analysis

Data were analysed thematically using Braun and Clarke’s six-step framework [27]. Coding was inductive, initially informed by the topic guide. Two researchers (IPN and PN) independently coded the transcript and resolved discrepancies through discussion. Preliminary themes were reviewed by the full research team to ensure accuracy and interpretive coherence. Coding continued until no new themes emerged, indicating thematic saturation. Data were managed in Excel. Anonymised quotes were used to illustrate key themes, and member checking was conducted by circulating the transcript to participants to confirm the accuracy of their statements, enhancing the credibility of the findings.

In line with COREQ criteria, member checking was conducted by sending the full transcript to all participants to verify the accuracy of their contributions. Four participants responded, confirming that the transcript accurately reflected their views, and no modifications were requested. This process enhanced the credibility and trustworthiness of the findings.

## 3. Results

The analysis of the focus group revealed three main themes, Table 1: (1) building a positive relationship with pregnant smokers, (2) pregnant smokers’ awareness of health risks, and (3) informative techniques used by smoking cessation advisors (SCAs). These themes were closely interconnected, with the relational approach influencing how information could be communicated and the extent to which smokers’ awareness impacted their motivation to quit.

### 3.1. Building a Positive Relationship with the Pregnant Smoker

Advisors consistently highlighted that discussions about smoking during pregnancy are inherently sensitive. There were mixed views and no fixed way of informing the smoker of the dangers of smoking while pregnant. The participants also believed it was important to find the balance of informing the pregnant women without being judgmental or overly assertive, as this may provoke a negative response or discourage the smoker:


*‘Because if you give them a hard time you run the risk of them never coming back and they’re expecting you to tell them off sort of thing. So, if you are nice to them about it there’s more chance that they will respond to you and carry on.’*
(SCA Nurse)

All advisors believed that they had to be empathetic and understanding; however, it was emphasised that due to the woman being pregnant, they only had limited time to conduct the consultations:


*‘…say I’ll think about it and I can come back in two- or three-weeks’ time or whatever and you can keep postponing it but knowing they’re pregnant you only have limited time and the sooner you do it the better, it is for the baby and for yourself.’*
(SCA midwife)

The participants believed it was important to focus mainly on the pregnant women, rather than on the unborn child. Therefore, the focus is placed on the pregnant woman’s own health and well-being, rather than framing her behaviour solely in terms of potential harm to the unborn child. *‘I always say that you are more important than the baby’* (SCA, midwife)

### 3.2. Pregnant Smoker’s Awareness of Health Risks

Advisors believed that pregnant smokers were already aware that smoking could harm the baby, as evidenced by their decision to attend the cessation sessions.: *‘With all that is in the media very few people do not know the risks of smoking and it is not the lack of knowledge but self-efficacy’* (SCA nurse). It was also deemed important to assess the level of understanding and knowledge that smokers have about the hazards:


*‘You try and gage what sort of level of understanding they are at and then say ‘well do you know it’s linked to cot death, respiratory problems for the baby and the problems in pregnancy are this’, and not try and scare them but just say the reality’.*
(SCA nurse)

Most advisors noted that, although their clients were aware of the risks, many did not necessarily want to hear detailed facts or statistics, or engage in in-depth discussions about the issue: *‘The responses that I have had are, ‘I Know, I know!’ as if they want to put an end to it. They don’t want to hear any more but they are others who do want to know but the majority do not want to know because they know what we are going to get at.*’ (SCA midwife)

Another key point highlighted by the SCAs was that some pregnant smokers, while aware of the harms of smoking, tended to minimise these risks by reflecting on their past experiences, thereby justifying their continued smoking. Many clients cited personal or family experiences to justify continued smoking, reflecting a gap between knowledge and motivation. This tension illustrates that awareness alone does not reliably lead to cessation. This contrast between awareness and minimisation reflects a psychological coping mechanism: while women recognise the dangers, they often rationalise continued smoking to manage guilt or stress. Such cognitive dissonance may explain why awareness alone does not translate into behavioural change.


*‘They would say that they smoked through their previous pregnancies or that their mother smoked through her pregnancy and it had not affected the baby. Others would say they know smoking can cause the baby to be born small, but that’s a positive in their mind! And then we have other who think that the stress of not smoking will cause more harm.’*
(SCA health psychologist)

Participants noted that, while most pregnant smokers had a general awareness of the harms of smoking to the unborn child, some lacked either awareness or specific knowledge of the associated risks. SCAs also observed that some women believed the stress of quitting might cause greater harm than continuing to smoke. As one midwife reflected, experts sometimes overestimate clients’ knowledge: *‘To some extent we assume they know more than they do because we work in this field; we do have this assumption*’. (SCA midwife)

Many participants suggested that their pregnant smoker clients were somewhat aware of the harms smoking posed to their own health but often lacked detailed knowledge of the specific effects: *‘I had this one girl who had no idea that she could run into health difficulties because she thought that was for old people.*’ (SCA nurse). The participants also suggested that they were able to pick up on cues from women who were either misled by information others had told them or simple were not aware of the exact dangers: *‘No, they just know it’s bad, just a general sea of badness but they don’t know actually know what emphysema means.’* (SCA nurse)

### 3.3. Informative Techniques Used by the SCA

The majority of the advisors did not want to emphasize the risks with the pregnant smoker as they did not want to overwhelm them: *‘To briefly skim over the risks and ask the smoker what they think the risks are and then to concentrate on helping them to stop smoking for themselves*’ (SCA midwife). They identified specific techniques for discussing these harmful effects, focusing on general health risks while making them personally relevant, such as impacts on appearance, rather than emphasising the effects on the unborn child:


*‘Sometimes I relay some of the heavy smokers I dealt with in my last study and I have emulated some of the things they have said to me to these young girls and have said this is what they have said about their teeth falling out and about how old they looked and they couldn’t breathe properly and I’ll say well you don’t have to go through that. That made a difference if I described individuals.’*
(SCA nurse)

A few participants described using shock tactics to capture clients’ attention, often through visual aids such as pictures of placentas, providing a concrete image for clients to reflect upon: *‘I have asked one of my ladies if she was going to look at her scan picture when she wants a cigarette and she said no she was just going to picture the placenta of her last baby because apparently she had seen a placenta and it had been awful. I think at that point they had probably said this is because you are a smoker’* (SCA midwife)

Similarly, others described using visual techniques through posters: *‘Majority know about babies being born small because when you ask them they will tell you yes my baby will be small but I have a couple of posters in my little corner with a baby in intensive care with tubes and that hoping that and seeing that these babies are premature or small for the size they are supposed to be.’* (SCA midwife)

Other, more direct techniques included offering pregnant smokers short videos—often advertisements illustrating the harms of smoking to the unborn child, as well as providing articles detailing these risks: *‘You might get better results of really spelling it out and so a few times I have done that and I think that’s quite effective as women really don’t know*’ (SCA nurse)

Participants emphasised the need to avoid overwhelming clients with information, recognizing that many women already felt guilty about smoking during pregnancy. Consequently, they carefully tailored discussions about the risks to the unborn baby, assessing each case individually to determine how much or whether to address the topic: *‘Depends on the women really, you have to weigh up because they are probably beating themselves up anyway about the fact that they relapsed. So, you have to kind of gage how destructive it would be to people, the guilt*’. (SCA midwife)

Overall, the advisors reported mixed perceptions of these techniques: visual aids and shock tactics occasionally captured attention, but some noted they could provoke guilt or overwhelm clients. The techniques were selectively applied based on individual client readiness and emotional state, emphasising the importance of tailoring interventions.

### 3.4. Integrative Interpretation

Across themes, two key patterns consistently emerged across all ten advisors. First, relational quality was foundational: trust, empathy, and non-judgmental communication enabled pregnant smokers to engage with cessation advice and behaviour-change strategies. Relational approaches influenced how informative techniques were received, allowing advisors to tailor discussions without provoking guilt or defensiveness. Second, a knowledge–motivation gap was evident: awareness of the harms of smoking alone did not reliably drive cessation. Advisors reported that addressing self-efficacy, emotional barriers, and cognitive rationalisations was essential to support quitting. Together, these patterns highlight that effective smoking cessation support depends on both strong relational engagement and targeted, individually tailored information, reinforcing the need for relationship-based interventions that account for psychosocial, emotional, and social barriers [8,9,10,11,12,13,14,15]. All participants’ perspectives were largely consistent, and no additional or contrasting themes emerged.

## 4. Discussion

The analysis of the focus group provided insight into the sensitivities and difficulties that advisors perceived when supporting pregnant women to stop smoking. The main themes identified were building a positive relationship between client and professional, assessing the level of awareness or lack of awareness of the health risks, and providing appropriate techniques and tactics to help them stop smoking. Although data were collected in 2012, the findings remain relevant. Persistent psychosocial and socioeconomic factors continue to influence maternal smoking, and gaps in cessation support for pregnant women are still evident [7,10,11,12,13,14,15,16]. Current guidance and training standards, including NICE (2021, updated 2025) and NCSCT (2023), continue to emphasise person-centred, tailored approaches, reflecting the ongoing applicability of the advisors’ experiences [20,23]. Data from this study may offer useful insight into how advisors perceived their work at the time.

Despite earlier studies suggesting that pregnant women are aware of the risks of smoking during pregnancy [12,13,14,15,16,17,18,19], advisors in this study perceived that women’s understanding of smoking-related risks varied. Advisors described assessing women’s level of knowledge about the harms of smoking in pregnancy, providing additional information when needed, and doing so in a way they felt was sensitive to women’s feelings and vulnerability. There was a consensus between advisors on the kinds of attitudes (e.g., ‘smoking is bad, but it won’t harm their baby’) and level of awareness the smokers had, though the specific techniques used to convey this information varied.

Previous studies report that many pregnant smokers acknowledge that smoking is unhealthy, but this awareness does not always translate into a belief that it will harm their unborn child [6]. Advisors perceived that many pregnant smokers did not hold strong views on the risks and consequences of smoking, which advisors felt might influence women’s motivation to maintain a smoke-free pregnancy, though this interpretation is based on advisor perceptions rather than direct evidence from pregnant women. These contradictions may reflect cognitive dissonance, where pregnant smokers rationalise continued smoking despite awareness of risks and stigma, which can lead to minimization of harm to reduce feelings of guilt or social judgement.

This study provides preliminary insight into how advisors approach discussions about smoking-related risks with pregnant women. Advisors reported confidence in providing tailored advice when appropriate, although this should not be taken as evidence that all health professionals are equally confident or effective in doing so [12]. Their confidence may partly reflect the intensive training, supervision, and structured protocols associated with the RCT environment, which may not mirror typical NHS practice. Their approach involved identifying smokers’ beliefs about health risks and using this understanding to raise awareness. Importantly, discussions of risk were often brief and selectively applied, reflecting an attempt to balance information provision with avoiding discouragement or disengagement.


**Practical Implications**


Previous literature has indicated that having a specific structure of information to inform pregnant women of the health effects would be beneficial; however, specific guidance is lacking [18,19]. These findings offer preliminary insight that a structured framework to guide discussions about health risks may be helpful to emphasize the potential benefits of quitting for the unborn child. In addition, the study provides preliminary insight into the information pregnant smokers may already have and highlights what additional guidance advisors might provide to support more informed decision-making about smoking cessation. These observations may complement existing NICE (2025) and NCSCT (2023) recommendations that emphasise tailoring interventions [17,20].


**Implications for research**


Further research is needed to enhance the effectiveness of informing pregnant women about the risks of second-hand smoke. Future research could examine whether the quality of rapport between professionals and clients influences how risk information is interpreted. Additionally, comparing quit rates between pregnant smokers who receive targeted educational interventions and those who do not could provide valuable insight into the impact of awareness programmes. Research should also further explore the socio-economic and psychological factors that sustain maternal smoking and hinder cessation among socially disadvantaged women, such as having a partner who smokes [4,12,16].


**Strengths and limitations**


A major strength of this study is the rich insight provided by Stop Smoking Advisors (SCAs) regarding the complexity of advising and motivating pregnant women to stop smoking. The findings are reinforced by the SCAs’ extensive experience across multiple hospitals in London and Kent, representing a diverse range of clinical settings and patient populations. This diversity may enhance the range of experiences captured, although findings remain specific to this small group. Additionally, involving one SCA in the analysis enhanced interpretive depth, aligning with principles of participatory research [27]. However, there are several limitations to this study: (1) a single focus group was used with a small sample, capturing only the perspectives of SCAs and not the pregnant women themselves, limiting generalisability. (2) Data were collected in 2012; although underlying behavioural and socio-economic challenges may remain relevant, policy, social context, or healthcare practice may have evolved. (3) All participants were female advisors; although this reflects the workforce profile in many maternity smoking cessation services, it may nevertheless limit the diversity of professional perspectives included. (4) The use of a group format may have introduced conformity pressures, potentially limiting the expression of dissenting views. (5) Advisors’ confidence and reported practices may have been shaped by the training and support provided within the RCT, which may not reflect routine NHS settings. Future research should expand the dataset, include multiple focus groups, and incorporate direct interviews with pregnant women to provide a more comprehensive understanding of cessation support dynamics.

## 5. Conclusions

This study achieved its aim of exploring SCAs’ perspectives on supporting pregnant women to stop smoking. The Findings highlight the importance of building trust, addressing cognitive and emotional barriers, and tailoring communication to individual knowledge, beliefs, and readiness to change. Advisors also noted a gap between general awareness of smoking risks and clients’ motivation to quit, emphasising the need to provide information without provoking guilt or disengagement.

The SCAs were drawn from multiple hospitals, representing diverse clinical settings and patient populations. These insights offer preliminary guidance for person-centred health promotion and structured risk discussions, but given the small sample, the findings remain exploratory. Further research with larger, more diverse populations, including input from pregnant smokers, would strengthen generalizability and inform more definitive practice recommendations.

## Figures and Tables

**Table 1 healthcare-13-03018-t001:** Integrated Themes, Sub-Themes, and Insights.

Main Theme (n/10)	Sub-Themes	Insight	Quote
Building positive relationships (10/10)	Balancing sensitivity and assertiveness; Focus on mother, not baby; Empathy within limited time	Relational quality is key: empathy and personalised attention increase receptiveness; directive approaches may provoke resistance	“Because if you give them a hard time…more chance they will respond.” (SCA Nurse)
Awareness of health risks (9/10)	Awareness of risks but low self-efficacy; Avoidance of facts; Minimising via personal/family experiences	Knowledge alone does not ensure behaviour change; clients rationalise risks or fear quitting stress more than smoking	“With all that is in the media very few people do not know the risks… it is not the lack of knowledge but self-efficacy.” (SCA Nurse)
Informative techniques (8/10)	Personalising via stories; Visual/shock tactics; Tailored info to avoid overwhelm	Tailoring content and intensity to emotional state is essential; relational quality moderates delivery	“Depends on the woman really…you have to gauge how destructive it would be.” (SCA Midwife)

## Data Availability

The data presented in this study are available on request from the corresponding author. Data contains personal information and cannot be shared publicly; access may be granted upon reasonable request.

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
