# Peer review of "Smoking Cessation Advisors’ Perspectives on Pregnant Women’s Attitudes on the Risks and Consequences of Smoking While Pregnant"

_healthcare, 2025, doi:10.3390/healthcare13233018_

Round 1

Reviewer 1 Report

Comments and Suggestions for Authors

The study is designed with an objective to investigate the experiences and opinions of smoking cessation advisors on pregnant smokers’ attitudes and beliefs about smoking while pregnant and the harm it may cause the unborn baby. The study is well drafted and creates interest among readers. However, the following are the points for improvement. 

  1. There is a need to consider common health complications associated with pregnancy, such as gestational diabetes mellitus, by considering https://doi.org/10.1080/03630242.2023.2197083,  DOI: 10.4103/2231-0738.199067,  doi: 10.5001/omj.2016.73 in the introduction to correlate smoking and pregnancy complications.
  2. Please provide the reason for selecting pregnant women between 10- and 24 weeks’ gestation who smoked five or more cigarettes a day before pregnancy and one or more during pregnancy. 
  3. What was the period of study, and who had collected information through face-to-face interaction?
  4. The criterion of inclusion and exclusion required to be clarified further. 
  5. Describe the statistical analysis used in this study
  6. Strengthen the discussion part of the article by having more external evidence for generalizing the outcome of the study.
  7. Please include DOI for each references

Author Response

Point-by-point response to Comments and Suggestions for Authors

1.There is a need to consider common health complications associated with

pregnancy, such as gestational diabetes mellitus, by considering

https://doi.org/10.1080/03630242.2023.2197083, DOI: 10.4103/2231-

0738.199067, doi: 10.5001/omj.2016.73 in the introduction to correlate smoking

and pregnancy complications.

We agree with this comment. Therefore, we have added discussion of common metabolic complications in pregnancy linked to smoking and included the suggested references. Alhtough most recent research states there is no link. This addition is located in the Introduction (lines 45–48).

‘In addition to these outcomes, smoking during pregnancy has been linked with metabolic complications such as gestational diabetes mellitus and impaired glucose tolerance [4]. However, recent meta-analysis found no clear association between maternal smoking during pregnancy and gestational diabetes (OR 1.06, 95% CI 0.95–1.19) [5].’

  1. Please provide the reason for selecting pregnant women between 10- and 24 weeks’

gestation who smoked five or more cigarettes a day before pregnancy and one or

more during pregnancy. We have clarified the rationale for participant selection. This is now detailed in the Methods section (lines 95–97):

“This gestational range was selected to ensure participants were recruited early enough in pregnancy for cessation to benefit foetal development, while the smoking thresholds identified women with established, regular smoking patterns suitable for behavioural intervention [23].

  1. What was the period of study, and who had collected information through face-to[1]face interaction? ]. We have added detailed information about the study period and data collection. This can be found in the Methods section (Lines 85 to 88 and Lines 92-93:

‘Participants were recruited using purpose sampling from the London Exercise and Pregnant Smokers (LEAP) randomised controlled trial (RCT). This trial was conducted between April 2009 and January 2014 and evaluated whether adding a physical-activity intervention (mainly walking on a treadmill) to standard cessation support improved quit rates among pregnant smokers [22,23]. ‘

‘In the main trial, face-to-face data collection with pregnant participants was conducted by these trained SCAs. The SCAs delivered weekly behavioural cessation sessions and recorded smoking status and intervention data at each visit.’

  1. The criterion of inclusion and exclusion required to be clarified further. We have included the criteria for both the main trial and for the focus group study lines 95 to 104:

‘This gestational range ensured enrolment early enough in pregnancy for smoking cessation to benefit fetal development. The smoking thresholds identified women with established, regular smoking patterns suitable for behavioural intervention [23]. Exclusion criteria included medical or obstetric complications that could limit physical activity, inability to complete questionnaires in English, substance dependence, or planned use of nicotine replacement therapy at baseline. …. For this study the inclusion criteria required that participants were actively providing or had recently provided smoking cessation support to pregnant women within NHS hospital or community maternity services and had completed NCSCT-accredited training. Exclusion criteria included advisors without direct client contact or those not trained to the national standard.’

  1. Describe the statistical analysis used in this study We have included the analysis for the main trial lines 99 to 100:

The main RCT employed intention-to-treat analysis with logistic regression to compare smoking cessation outcomes between intervention groups [23].

  1. Strengthen the discussion part of the article by having more external evidence for

generalizing the outcome of the study.

We have strengthened the Discussion (lines 236–301) by adding recent external evidence and linking our findings to international literature, including Vila-Fariñas et al. (2024), Fletcher et al. (2022, 2025), Tatton & Lloyd (2023), David et al. (2024), Khanal et al. (2023), and NICE/NCSCT guidance (2023–2025). We have also expanded the practical and research implications and balanced the strengths and limitations to support generalizability. For example,

‘Although data were collected in 2012, the findings remain relevant. Persistent psychosocial and socioeconomic factors continue to influence maternal smoking, and gaps in cessation support for pregnant women are still evident [7,10–16]. Current guidance and training standards, including NICE (2021, updated 2025) and NCSCT (2023), continue to emphasise person-centred, tailored approaches, reflecting the ongoing applicability of the advisors’ experiences [20,23]. Thus, these data provide enduring insight into challenges and strategies in supporting pregnant women to quit smoking’. This is also stated in the practical implementation and implications for research sections.’

  1. Please include DOI for each reference

All DOI for each reference have now been included.

We believe these revisions fully address the reviewers’ feedback and significantly improve the manuscript’s quality, clarity, and contemporary relevance. Thank you for considering our revised submission for publication.

Sincerely,
Dr Iliatha Papachristou Nadal
(on behalf of all co-authors)
King’s College London
Email: iliatha.papachristounadal@kcl.ac.uk

Reviewer 2 Report

Comments and Suggestions for Authors

Abstract (lines 16–29)

  • Lines 22–23: Merely listing the three themes is vague. Add a brief description of what advisors actually reported under each theme.
  • Lines 25–26: Practice implications are framed too prescriptively given the small, single focus group. Please soften claims (“may help” instead of “can help”).
  • Missing: A brief acknowledgment of limitations (small sample, single focus group) would increase transparency.

Introduction (lines 31–71)

  • Lines 32–39: The harms of smoking are detailed extensively but repetitively. This section could be condensed into 2–3 sentences.
  • Lines 41–49: The summary of psychosocial barriers is relevant but reads like a list. Please synthesize into a more straightforward narrative (e.g., “although women acknowledge risks, barriers such as dependence, stress, and social influences impede cessation”).
  • Line 63–65: The statement that midwives and professionals lack training/tools is crucial for your rationale. Consider moving this earlier to highlight the knowledge gap sooner.
  • Lines 69–71: The study aim is stated, but could be more concise and focused.  
  • Missing: Integration of recent literature (post-2020) is needed to establish relevance.

Methods (lines 73–135)

  • Lines 74–83: Clarify the sampling strategy—was it purposive, convenience, or limited to RCT participation? Without this, reproducibility is limited.
  • Line 119–121: The moderator was also the PI of the RCT, which risks bias. This must be acknowledged as a limitation.
  • Lines 109–118: The topic guide is summarized but not fully presented. Including the complete guide in an appendix would strengthen transparency.
  • Lines 126–133: Thematic analysis is appropriate, but more detail is required on coding (number of coders, disagreement resolution, evidence of saturation). Noting that the use of Excel alone is insufficient to demonstrate rigor.
  • Lines 134–135: Member checking (transcripts returned) is a strength—consider highlighting this more clearly.

Results (lines 137–235)

  • Line 141 (Table 1): The table is clear but basic. Please indicate theme prevalence (e.g., number of advisors mentioning each theme) and link to representative quotes.
  • Lines 145–163: The “rapport” theme is well-illustrated, but please clarify whether all advisors agreed on this or if differences existed between roles (nurses, midwives, psychologists).
  • Lines 165–194: This section shows both awareness and denial/minimization of risks. Please synthesize these contradictions explicitly rather than leaving them side by side.
  • Lines 203–235: Techniques (shock tactics, posters) are interesting, but you should address whether advisors perceived these as effective or problematic.
  • Missing: A visual model (diagram) linking the three themes would improve clarity and presentation.

Discussion (lines 236–301)

  • Lines 236–247: The discussion repeats findings without deeper interpretation. Please expand by connecting the contradictions in smokers’ awareness/minimization to theory (e.g., stigma, cognitive dissonance).
  • Lines 248–253: The comparison with previous studies is valid but outdated. More recent literature (2020–2024) should be integrated.
  • Lines 254–263: Advisor confidence is interesting but may reflect their RCT training—please clarify.
  • Lines 266–271: Practical implications are overstated. Reframe as exploratory insights, not definitive recommendations.
  • Lines 283–293: Strengths/limitations are acknowledged, but key issues are missing:
    • Dataset age (2012, published 2025).
    • Moderator’s dual role as PI.
    • Limited generalisability (10 SCAs, London/Kent only).
  • Lines 299–300: The reference to the “current economic recession” is speculative and not supported by data—please remove or reframe.

Conclusions (lines 295–301)

  • The conclusions align with your aims but remain too general. They should explicitly reflect whether the aim (to explore SCAs’ perspectives) was achieved and what novel insights were gained.
  • Practice recommendations should be softened, given the small qualitative sample.
Comments on the Quality of English Language

The English could be clearer to present the research better. Some sentences are rambling and repetitive, especially in the Introduction and Discussion sections. Editing for conciseness and accuracy would improve the manuscript.

Author Response

Response to Reviewers

Dear Editor,

We would like to thank you and the reviewers for your constructive feedback on our manuscript titled: “Smoking cessation advisors’ perspectives on pregnant women’s attitudes on the risks and consequences of smoking while pregnant.”

We greatly appreciate the opportunity to revise and resubmit our paper. The reviewers’ comments were insightful and have substantially strengthened the clarity, methodological transparency, and relevance of our work. Below, we outline the key revisions and how each comment has been addressed.

  1. 3. Point-by-point response to Comments and Suggestions for Authors

    Point 1 to 3: Abstract (lines 16–29)

    • Lines 22–23: Merely listing the three themes is vague. Add a brief description of what advisors actually reported under each theme.

    • Lines 25–26: Practice implications are framed too prescriptively given the small, single focus group. Please soften claims (“may help” instead of “can help”).

    • Missing: A brief acknowledgment of limitations (small sample, single focus group) would increase transparency.

    Thank you for suggesting these three changes. We agree with this comment. Therefore, we have edited the abstract to address these points a) adding a sentence after the main these, tempering the language to include may and can and highlighting the small focus group.  While based on a single focus group, Findings suggest that while many pregnant smokers recognise the health risks, this awareness alone may not lead to cessation. Conclusions Adopting a person-centred approach that considers pregnant smokers’ knowledge, beliefs, and emotional readiness may help advisors deliver timely and appropriate information to encourage cessation. Practice implications could benefit from a more structured yet flexible framework to guide sensitive discussions about smoking risks and to reinforce the potential benefits of quitting for both maternal and fetal wellbeing.

    Points 4 to 8 Introduction (lines 31–71) We have, accordingly, revised to emphasize these points.

    • Lines 32–39: The harms of smoking are detailed extensively but repetitively. This section could be condensed into 2–3 sentences. Much of the introduction is now condensed. Such as the harms of smoking – lines 43 to 47. ‘Maternal smoking remains one of the leading preventable causes of adverse pregnancy outcomes [1]. Recent meta-analytic and national cohort studies confirm that prenatal smoking increases the risk of miscarriage and preterm birth by 30–40%, doubles the likelihood of low birth weight, and raises the risk of stillbirth by about 50% [2,3].’

    • Lines 41–49: The summary of psychosocial barriers is relevant but reads like a list. Please synthesize into a more straightforward narrative (e.g., “although women acknowledge risks, barriers such as dependence, stress, and social influences impede cessation”). Also refined is ‘the list style’ paragraph; lines 57-60: ‘Although most pregnant women recognize the harms of smoking, psychological dependence, social pressures, stigma, and the tendency to normalize risk often impede cessation. Women who successfully quit tend to hold stronger beliefs about smoking’s harms, highlighting the importance of interventions that address both cognitive and emotional barriers [9,10–15,18].’

    • Line 63–65: The statement that midwives and professionals lack training/tools is crucial for your rationale. Consider moving this earlier to highlight the knowledge gap sooner. Now that the introduction has been refined and condensed the healthcare professional section is directly after the psychosocial barriers paragraph. Lines 62 to 71.

    • Lines 69–71: The study aim is stated, but could be more concise and focused. The aim has been condensed to ‘This study explored smoking cessation advisors’ (SCA) perspectives on pregnant women’s attitudes toward smoking and perceived risks to the unborn child.’ Lines 78-79.

    • Missing: Integration of recent literature (post-2020) is needed to establish relevance. References have been updated, including policy and training guidelines, systematic review and UK-based evidence-based research. As evidence in the reference list.

    Methods (lines 73–135)

    • Lines 74–83: Clarify the sampling strategy—was it purposive, convenience, or limited to RCT participation? Without this, reproducibility is limited. Lines 84-85 has now included a statement on sampling: ‘Participants were recruited using purpose sampling from the London Exercise and Pregnant Smokers (LEAP) randomised controlled trial (RCT).’

    • Line 119–121: The moderator was also the PI of the RCT, which risks bias. This must be acknowledged as a limitation. We have included a statement regarding PI involvement as facilitator. lines 142-144 –‘The moderator (last author), who also served as the principal investigator of the main trial, facilitated the focus group. While their broader involvement in the project could have introduced bias, they were not directly engaged in participant support or smoking cessation delivery, thereby reducing the likelihood of influencing responses.’

    • Lines 109–118: The topic guide is summarized but not fully presented. Including the complete guide in an appendix would strengthen transparency. We have now included the full topic guide as an appendix (1).

    • Lines 126–133: Thematic analysis is appropriate, but more detail is required on coding (number of coders, disagreement resolution, evidence of saturation). Noting that the use of Excel alone is insufficient to demonstrate rigor. We have expanded the analysis section to include number of coders, consensus, and saturation statement. Lines 151 to 155. Data were analysed thematically using Braun and Clarke’s six-step framework [27]. Coding was inductive, initially informed by the topic guide. Two researchers (IPN and PN) independently coded the transcript and resolved discrepancies through discussion. Preliminary themes were reviewed by the full research team to ensure accuracy and interpretive coherence. Coding continued until no new themes emerged, indicating thematic saturation. Data were managed in Excel..’

    • Lines 134–135: Member checking (transcripts returned) is a strength—consider highlighting this more clearly. Results (lines 137–235) A statement on member checking as been included line 155 to 156:  ‘Anonymised quotes were used to illustrate key themes, and member checking was conducted by circulating the transcript to participants to confirm the accuracy of their statements, enhancing the credibility of the findings’

    • Line 141 (Table 1): The table is clear but basic. Please indicate theme prevalence (e.g., number of advisors mentioning each theme) and link to representative quotes. A revised table has been included to state number of advisors in each them and quotes. (page 4/5)

    • Lines 145–163: The “rapport” theme is well-illustrated, but please clarify whether all advisors agreed on this or if differences existed between roles (nurses, midwives, psychologists). As above

    • Lines 165–194: This section shows both awareness and denial/minimization of risks. Please synthesize these contradictions explicitly rather than leaving them side by side. We have synthesised these two concepts by summaries lines 202 to 206: ‘Many clients cited personal or family experiences to justify continued smoking, reflecting a gap between knowledge and motivation. This tension illustrates that awareness alone does not reliably lead to cessation. This contrast between awareness and minimisation reflects a psychological coping mechanism: while women recognise the dangers, they often rationalise continued smoking to manage guilt or stress. Such cognitive dissonance may explain why awareness alone does not translate into behavioural change.’

    • Lines 203–235: Techniques (shock tactics, posters) are interesting, but you should address whether advisors perceived these as effective or problematic. Address by adding statement at the bottom of this theme: lines 257 - 261

    ‘Overall, the advisors reported mixed perceptions of these techniques: visual aids and shock tactics occasionally captured attention, but some noted they could provoke guilt or overwhelm clients. The techniques were selectively applied based on individual client readiness and emotional state, emphasising the importance of tailoring interventions’

    2 • Missing: A visual model (diagram) linking the three themes would improve clarity and presentation. Rather than a visual model we have added a section lines 262 to 270 on the integrative interpretation to discuss patterns across themes.

    3.4. Integrative Interpretation

    Across themes, two key patterns consistently emerged across all ten advisors. First, relational quality was foundational: trust, empathy, and non-judgmental communication enabled pregnant smokers to engage with cessation advice and behaviour-change strategies. Relational approaches influenced how informative techniques were received, allowing advisors to tailor discussions without provoking guilt or defensiveness. Second, a knowledge–motivation gap was evident: awareness of the harms of smoking alone did not reliably drive cessation. Advisors reported that addressing self-efficacy, emotional barriers, and cognitive rationalisations was essential to support quitting. Together, these patterns highlight that effective smoking cessation support depends on both strong relational engagement and targeted, individually tailored information, reinforcing the need for relationship-based interventions that account for psychosocial, emotional, and social barriers [8–15]. All participants’ perspectives were largely consistent, and no additional or contrasting themes emerged.’

    Discussion (lines 236–301)

    • Lines 236–247: The discussion repeats findings without deeper interpretation. Please expand by connecting the contradictions in smokers’ awareness/minimization to theory (e.g., stigma, cognitive dissonance). We have added theoretical interpretation:

    ‘These contradictions may reflect cognitive dissonance, where pregnant smokers rationalize continued smoking despite awareness of risks, and stigma, which can lead to minimization of harm to reduce feelings of guilt or social judgment.’ linking awareness–denial patterns to behavioural theory.

    • Lines 248–253: The comparison with previous studies is valid but outdated. More recent literature (2020–2024) should be integrated. We integrated current evidence, e.g. NICE (2021, updated 2025), NCSCT (2023), and recent studies [7, 10–16], demonstrating continuing relevance.

    • Lines 254–263: Advisor confidence is interesting but may reflect their RCT training—please clarify.

    We tempered this by adding, e.g. “although this should not be taken as evidence that all professionals are equally confident or effective,” acknowledging context and limits.

     • Lines 266–271: Practical implications are overstated. Reframe as exploratory insights, not definitive recommendations. We softened language, e.g. “findings suggest,” “provide preliminary insight,” “may reflect,” positioning results as exploratory.

    • Lines 283–293: Strengths/limitations are acknowledged, but key issues are missing: o Dataset age (2012, published 2025).

    o Moderator’s dual role as PI.

    o Limited generalisability (10 SCAs, London/Kent only).

    We have added discussion of these factors. For example:
    ‘Data were collected in 2012; although underlying behavioural and socio-economic challenges may remain relevant, policy, social context, or healthcare practice may have evolved.”
    We also acknowledged the moderator’s dual role: “Although the moderator facilitated the discussions, they were not directly involved in advising participants, supporting impartiality while also serving as PI of the RCT.’
    Limited generalisability was addressed by noting that SCAs were drawn from multiple hospitals across London and Kent, representing diverse clinical settings, but that findings may not capture all regional variations.

    • Lines 299–300: The reference to the “current economic recession” is speculative and not supported by data—please remove or reframe. All mention of economic recession has been removed from the discussion to avoid unsupported statements.

    Conclusions (lines 295–301)

    • The conclusions align with your aims but remain too general. They should explicitly reflect whether the aim (to explore SCAs’ perspectives) was achieved and what novel insights were gained. We have clarified that the study achieved its aim and highlighted novel insights. For example:
    ‘This study achieved its aim of exploring SCAs’ perspectives on supporting pregnant women to stop smoking. Findings highlight the importance of building trust, addressing cognitive and emotional barriers, and tailoring communication to individual knowledge, beliefs, and readiness to change.’

    • Practice recommendations should be softened, given the small qualitative sample.

    We have revised practice recommendations to indicate preliminary guidance rather than definitive advice. For example:
    ‘These insights offer preliminary guidance for person-centred health promotion and structured risk discussions, but given the small sample, findings remain exploratory. Further research with larger, more diverse populations, including input from pregnant smokers, would strengthen generalisability and inform more definitive practice recommendations.’

We believe these revisions fully address the reviewers’ feedback and significantly improve the manuscript’s quality, clarity, and contemporary relevance. Thank you for considering our revised submission for publication.

Sincerely,
Dr Iliatha Papachristou Nadal
(on behalf of all co-authors)
King’s College London
Email: iliatha.papachristounadal@kcl.ac.uk

Reviewer 3 Report

Comments and Suggestions for Authors

Author Response

For review article

  1. Title – please delete “Title” in the beginning and full stop at the end of this title.

The word title has now been deleted

  1. Line 33: “Is it also known that…” Would this sentence be better written as: It is also known that….
  2. This has been changed to ‘Smoking while pregnant is known to increase infant mortality, morbidity…’
  3. Thank you for providing the COREQ checklist. However, the submitted form is currently blank and does not indicate where each of the 32 items is addressed in the manuscript. Please complete the checklist by specifying the page number/section where each item is reported.

COREQ checklist is now complete and included as an appendix

  1. COREQ item 22: Data saturation – It is unclear whether data saturation was achieved or discussed in the manuscript. Please clarify whether data saturation was considered during data collection or analysis. If it was achieved, indicate how this was determined (e.g., no new codes or themes emerging). If not discussed, please consider adding a brief statement to address this point.

 A sentence has been added to the data analysis section line 153: Coding continued until no new codes or themes emerged, indicating that thematic saturation was achieved.

  1. COREQ item 32: Is there a description of diverse cases or minor themes? If so, please elaborate further in the manuscript. If not, please state this explicitly in the Results section to indicate that no additional or contrasting themes were identified.

A sentence has been added to the results section under the interpretation section ‘All participants’ perspectives were largely consistent, and no additional or contrasting themes emerged.’ Line 268

  1. While the authors note that the data were collected in 2012 and have provided some justification for their relevance, this explanation could be elaborated further. Expanding the discussion on why these data remain meaningful in the current context (2025), for example, by linking to data limited on advisors’ perspectives on smoking among pregnant and/or persistent challenges in maternal health and smoking cessation, would strengthen the manuscript’s credibility and highlight its continuing contribution to the field. Other viewpoints would be possible. We have include a section in the discussion to address this point: line 274 : ‘Although data were collected in 2012, the findings remain relevant. Persistent psychosocial and socioeconomic factors continue to influence maternal smoking, and gaps in cessation support for pregnant women are still evident [7,10–16]. Current guidance and training standards, including NICE (2021, updated 2025) and NCSCT (2023), continue to emphasise person-centred, tailored approaches, reflecting the ongoing applicability of the advisors’ experiences [20,23]. Thus, these data provide enduring insight into challenges and strategies in supporting pregnant women to quit smoking.’

  1. It appears that only one focus group discussion was conducted. Could the authors please confirm whether this is correct? If so, please discuss how conducting a single group might have affected the depth or breadth of the data, and whether data saturation was achieved or considered. Providing a brief justification (e.g., exploratory design, participant accessibility, or contextual constraints) would help strengthen the methodological transparency of the study

We confirm that only one focus group was conducted. This is addressed in the Methods section under Data Analysis, where we note that thematic saturation was achieved and justify the exploratory design.

We believe these revisions fully address the reviewers’ feedback and significantly improve the manuscript’s quality, clarity, and contemporary relevance. Thank you for considering our revised submission for publication.

Sincerely,
Dr Iliatha Papachristou Nadal
(on behalf of all co-authors)
King’s College London
Email: iliatha.papachristounadal@kcl.ac.uk

Round 2

Reviewer 2 Report

Comments and Suggestions for Authors

The authors have addressed the majority of previously raised issues. Below are my observations on the revision quality and suggestions for further strengthening the manuscript.

- Introduction: redundancy persists between paragraphs 1–3 in Version 2. The statement “financial stress and social disadvantage continue to shape smoking behaviours” should include a citation directly after the claim for precision.

= Methods: The aythors mention transcript confirmation, but not whether any participants responded, nor whether their feedback led to changes. COREQ encourages reporting this explicitly.

- Discussion:

  • The revised text reasonably tempers claims, but should still briefly clarify if the confidence described may stem from the intensive RCT training context, not typical practice.
  • Avoid overinterpretation: although the prescriptive tone was softened, some sentences still verge on generalising beyond the sample, e.g.:
    • “These insights offer preliminary guidance…” → acceptable
    • but some lines still imply broader applicability than warranted by a single focus group.

- Strengths and Limitations: Acknowledgement that all advisors were female (which may shape rapport, communication styles, and perceptions). A brief note that the group setting may have introduced conformity pressures (a COREQ-relevant point).

= Supplementary Files: COREQ checklist is partially complete; consider filling in item 28 (“participant feedback on findings”) more explicitly.

Comments on the Quality of English Language

Several sentences—particularly in the Introduction, Abstract, and Discussion—would benefit from minor language polishing to ensure precision, remove residual redundancy, and improve flow. 

Author Response

Cover Letter: Response to Reviewer Feedback

Dear Editor,

Thank you for the opportunity to revise our manuscript, “Smoking cessation advisors’ perspectives on pregnant women’s attitudes on the risks and consequences of smoking while pregnant.” We are grateful to the reviewers for their thoughtful and constructive comments, which have helped us strengthen the clarity, rigour, and transparency of the paper.

We are pleased to confirm that all outstanding issues raised in the most recent round of review have now been fully addressed. The manuscript has been carefully revised, and key changes have been clearly highlighted throughout the text.

Summary of Revisions

  1. Introduction – removal of redundancy and clarification of citations
    • Redundant material between the first three paragraphs has been streamlined.
    • A citation has been added directly after the statement regarding financial stress and social disadvantage to improve precision and alignment with reviewer guidance.
  2. Methods – clarification of transcript confirmation and participant feedback
    • We now explicitly state how many participants responded during member checking (four participants) and confirm that no changes were requested.
    • This information has been added both to the Data Analysis section and the COREQ checklist (Supplementary File 2, Item 28).
  3. Discussion – addressing remaining concerns about interpretation and context
    • Statements potentially implying broader generalisability have been further softened.
    • We have clarified that advisors’ reported confidence may reflect the intensive training and structured support associated with the RCT context, which may differ from routine NHS practice.
    • Additional adjustments were made to ensure no prescriptive statements or unwarranted extrapolations beyond the small, context-specific sample.
  4. Strengths and Limitations – COREQ-aligned transparency
    • The section now includes the reviewer-requested reflections on: the all-female sample of advisors, the potential influence of group dynamics and conformity pressures in a focus group format, and the impact of RCT-specific training on advisors’ confidence and practices.
    • These additions enhance completeness and methodological transparency in line with COREQ guidance.
  5. Supplementary Files – COREQ checklist
    • Item 28 (‘participant feedback on findings’) has been completed more explicitly to reflect the member-checking process.

We believe these revisions have further strengthened the manuscript and ensured full consistency with COREQ and the journal’s standards for qualitative reporting.

We appreciate the reviewers’ constructive engagement with our work and hope that the revised manuscript meets all remaining expectations.

Thank you for considering this revised submission. We look forward to your response.

With kind regards,

Dr Iliatha Papachristou Nadal

King’s College London